# Photochemical Printing of Plasmonically Active Silver Nanostructures

**DOI:** 10.3390/ijms21062006

**Published:** 2020-03-16

**Authors:** Marcin Szalkowski, Karolina Sulowska, Martin Jönsson-Niedziółka, Kamil Wiwatowski, Joanna Niedziółka-Jönsson, Sebastian Maćkowski, Dawid Piątkowski

**Affiliations:** 1Institute of Physics, Faculty of Physics, Astronomy and Informatics, Nicolaus Copernicus University in Toruń, Grudziadzka 5, 87-100 Torun, Poland; marszal@fizyka.umk.pl (M.S.); sulowska@doktorant.umk.pl (K.S.); kamilw@doktorant.umk.pl (K.W.); dapi@fizyka.umk.pl (D.P.); 2Institute of Low Temperature and Structure Research, Polish Academy of Sciences, Okolna 2, 50-422 Wrocław, Poland; 3Institute of Physical Chemistry, Polish Academy of Sciences, Kasprzaka 44/52, 01-224 Warszawa, Poland; martinj@ichf.edu.pl (M.J.-N.); jniedziolka@ichf.edu.pl (J.N.-J.)

**Keywords:** silver nanostructures, silver islands film, silver deposition, metal enhanced luminescence

## Abstract

In this paper, we demonstrate plasmonic substrates prepared on demand, using a straightforward technique, based on laser-induced photochemical reduction of silver compounds on a glass substrate. Importantly, the presented technique does not impose any restrictions regarding the shape and length of the metallic pattern. Plasmonic interactions have been probed using both Stokes and anti-Stokes types of emitters that served as photoluminescence probes. For both cases, we observed a pronounced increase of the photoluminescence intensity for emitters deposited on silver patterns. By studying the absorption and emission dynamics, we identified the mechanisms responsible for emission enhancement and the position of the plasmonic resonance.

## 1. Introduction

Metallic nanostructures can modify the spectral properties of quantum emitters localized in their vicinity. When both the distance and the spectral relations between them are properly chosen, absorption and emission rates can be significantly enhanced [1,2,3]. Modern chemical synthesis methods enable to control and tailor the spectral properties of the metallic nanoparticles. By changing sizes and shapes of nanoparticles (spheres, rods, triangles, stars, rings, wires, etc.), spectral positions of the resonance peak can be shifted across a broad spectral range, from ultraviolet to near-infrared [4,5]. The position of the plasmon resonance can be additionally controlled by changing the type of material that the particle is made of [6]. Presently, noble metals like silver and gold are the most widely used. Since metallic nanoparticles are usually synthesized using wet chemistry techniques, their optical properties are inevitably affected by particle size/shape dispersion. Thus, the optical properties of a particular nanoparticle can only be investigated using advanced experimental techniques, based on single-molecule detection [7], and in the case of macroscopic experiments, only a statistically averaged response from the sample can be probed.

Among plasmonically active platforms, silver islands films (SIFs) play an important role. The SIF substrate consists of randomly deposited silver islands, formed during a reduction reaction of silver nitrate on glass [8,9]. The synthesis is quite straightforward and low-cost, and such substrates have been shown to exhibit strong plasmonic properties. Namely, they can modify transition rates of nearby emitters. It has been shown, for instance, that quantum dots deposited on SIFs feature emission enhancement factors of about 5 [10], whereas even 200-fold fluorescence enhancement has been recently observed for photosynthetic complexes deposited on SIFs [11]. Besides fluorescence microscopy, SIFs have also been applied to increase the sensitivity of Raman [12] and TIRF microscopy [13]. Nonetheless, with respect to the metallic nanoparticles, SIFs feature some disadvantages. First of all, due to a high degree of sample inhomogeneity, the spatial distribution of islands as well as their sizes and shapes are random and cannot be controlled. In addition, the method of SIF fabrication implies in principle that the whole substrate is covered with silver islands, limiting their use as high optical contrast substrates. 

In this work, we demonstrate that silver nanostructures fabricated using a laser-induced photochemical reaction [14,15] are plasmonically active for both Stokes and anti-Stokes emitters. The method, in contrast to the wet chemistry SIF preparation, uses a tightly focused laser beam to initialize the reaction. Depending on the resolution of the optical system, the paths can feature submicrometer widths with practically no restrictions regarding their shapes and sizes. Scanning electron microscopy images confirm island-like morphology of the paths. The results of fluorescence microscopy on the other hand, indicate that these silver nanostructures can enhance the optical response of two qualitatively different emitters: peridinin-chlorophyll-proteins (PCPs) and rare-earths doped nanocrystals (NCs). PCP complexes are efficient natural proteins responsible for efficient light harvesting complexes [16,17], while the up-converting NCs allow conversion of the light from the infrared to the visible [18]. Experimentally measured enhancements are of the order of 5–7, and both an increase of absorption as well as radiative emission rates contribute to this remarkable effect. The demonstration that such high quality plasmonically-active patterns of arbitrary shape can be used for enhancing the optical response of organic and inorganic materials opens a way to implement these structures for optoelectronic and sensing devices. 

## 2. Results and Discussion

The influence of silver islands, fabricated using the photochemical approach on the optical properties of emitters, was evaluated for both Stokes and anti-Stokes nanostructures, i.e., photoactive protein, peridinin-chlorophyll-protein, and rare-earth doped nanocrystals, respectively. 

### 2.1. Photoactive Protein

Peridinin-chlorophyll-protein is a well-known light-harvesting complex. It consists of eight optically active peridinin molecules and two chlorophylls [19]. Its absorption spectrum is dominated by a broad band localized between 400 and 550 nm, attributed to the electric-dipole transition in peridinins and absorption of the chlorophylls via the Soret band (Figure 1a). Additional absorption at 655 nm comes from chlorophylls. The emission can be activated either via peridinins, which transfer the excitation energy to the chlorophylls, or via chlorophyll absorption bands. The Stokes-shifted emission of the PCP is observed at 670 nm and it is characterized by a monoexponential decay profile, as shown in Figure 1b. 

The sample was prepared by spin-coating (3000 rpm for 1 min) the aqueous solution of PCP (2 μg/mL) on a glass substrate with previously deposited silver stripes. Experiments were carried out using wide-field fluorescence microscopy and confocal fluorescence microscopy for determining the intensities and decay curves of PCP emission, respectively. The wide-field microscope (Eclipse Ti-S, Nikon, Japan) was equipped with an oil immersion objective (Plan Apo 60× NA = 1.4, Nikon, Japan), a set of LED illuminators, and an EMCCD camera (iXon, Andor, UK) mounted in the detection channel. The confocal microscope used the same objective, picosecond excitation lasers (BDL-SMN series, Becker&Hickl, Germany), and fast detection based on time correlated single photon counting system (SPC-130-EMN, Becker&Hickl, Germany) coupled with single photon counting module (ID100, ID Quantique, Switzerland).

Emission intensity maps obtained for four excitation wavelengths of 405 nm, 480 nm, 535 nm, and 630 nm, are displayed in Figure 2. The broad absorption spectrum of PCP complexes makes it possible to use a range of excitation wavelengths, an approach often applied for studying plasmonic interactions in similar systems [20]. The sequence of excitation wavelengths was reversed (from 630 nm to 405 nm) in order to minimize any influence of photobleaching, which is expected to be the strongest for shorter wavelengths. The fluorescence intensity maps indicate homogeneous distribution of the PCP complexes across the surface, except for one agglomerate. There were two important observations: (1) the emission intensity of PCP complexes placed in the vicinity of the silver stripes was considerably higher than elsewhere, and (2) the scale of this enhancement depended on the excitation wavelength. Both results confirm that silver islands fabricated using the photochemical approach not only can be printed on the surface with remarkable accuracy, but also that these structures are plasmonically active. In other words, they can be used for controlling the optical properties of emitters located at their proximity. Calculated enhancement factors are equal to approximately 2 for 405 nm, 3 for 480 nm, 2.5 for 535 nm, and around 1.5 for 630 nm. It is known that spatial localization of the electromagnetic field by metallic nanoparticles may result in plasmon enhanced absorption in nearby emitters [21]. Indeed, the results of fluorescence imaging indicate that photochemically deposited silver stripes enhance the PCP absorption, and that this effect is wavelength-dependent, with the maximum enhancement detected for 480 nm, which corresponds approximately to the plasmon resonance.

In addition to enhancing the absorption, metallic nanostructures may also—under appropriate geometrical and spectral conditions—influence the radiative properties of the emitters. In order to probe this process, we compared fluorescence transients measured of PCP complexes deposited on silver stripes and on glass. For the excitation, we used picosecond lasers operating at 405 nm, 488 nm, and 630 nm. The emission was extracted using a bandpass filter (670/10 nm), matching the emission of PCP chlorophylls. The comparison plotted in Figure 1b confirms that interaction with silver stripes indeed affect the emission of chlorophyll in the protein. In contrast to the reference, the decay measured for PCPs on silver stripes consists of two components: a fast one of about 0.3 ns, and a slow one of about 2.8 ns, referring to the emission from isolated PCPs. The presence of the fast component indicates an increase of the emission rates due to the interaction with the silver stripes, known as Purcell effect [22]. In this process, the probability of spontaneous emission is increased due to modified local density of states, caused by the presence of the metallic nanoparticles [23]. Higher probability of the spontaneous emission leads to more absorption–emission cycles realized in a second by a single emitter, and thus contributes to the emission enhancement. The slow component, however, indicates that some PCP complexes are not coupled to the silver stripe—probably because of larger separation from the stripe—and decay with the rate of the reference. We found no measurable dependence of the emission dynamics on the excitation wavelength.

### 2.2. Up-Converting Nanocrystals

Erbium-ytterbium doped NaYF_4_ nanocrystals were synthesized by wet chemistry, as described elsewhere [24]. They exhibit efficient anti-Stokes luminescence, commonly referred to as energy transfer up-conversion emission [25]. In this system, Yb^3+^ plays the role of a donor, whereas Er^3+^ is the acceptor of the energy. The mechanism of the energy conversion involves several electronic states of erbium and is presented in Figure 3. Nanocrystals excited at 980 nm show visible up-conversion emission lines, centered at 550 and 660 nm, assigned to the ^2^H_11/2_+^4^S_3/2_→^4^I_15/2_ and ^4^F_9/2_→^4^I_15/2_ electric-dipole transitions in Er^3+^ ions, respectively (Figure 4a) [25]. Typical photoluminescence transient acquired at 660 nm for a single NC on glass is presented in Figure 4b. The transient has a bi-exponential character, and can be globally described by the averaged time constant of about 120 μs [25].

Nanocrystals, with an average diameter of about 30 nm, were dispersed in chloroform, where they form an optically stable colloid. A quantity of 10 μL of the NCs colloid (10 mg/mL) was spin-coated (3000 rpm for 1 min) on a glass substrate with previously fabricated silver stripes, as presented in Figure 5a. The concentration of the colloid was optimized to obtain a close to homogeneous layer of the nanocrystals on the sample surface. To fabricate the metallic nanostructure, we applied the same protocol of silver deposition; however, the exposure time was in this the same for each stripe (200 ms). For measuring luminescence of NCs, we used a confocal microscope (Eclipse Ti-S, Nikon, Japan) equipped with an oil immersion objective (Plan Apo 60× NA = 1.4, Nikon, Japan), a 10 mW excitation laser operating at 980 nm, and a single photon counting module (SPCM-AQR-16, Perkin Elmer, Canada). As the up-conversion is a more power-demanding process, we used a confocal excitation and detection scheme in order to assure for sufficient power densities. Photoluminescence intensity map acquired for a 660 nm emission is displayed in Figure 5b. The NCs are distributed rather homogeneously on the substrate, and similarly as for the PCP complexes, NCs placed in the vicinity of the silver stripes feature a much stronger emission compared with the reference. Indeed, there are some very bright emission spots, which we attribute to the emission originating from nanocrystals placed in the optimal positions with respect to the silver islands. In this case, the enhancement factor reaches values of about 6. The intensity of the up-conversion emission depends quadratically on the excitation power. Thus, if enhanced absorption appears, it leads to a greater enhancement factor than observed for Stokes-like emitters. Importantly, the emission pattern recorded for the second emission band of the NCs (550 nm) is identical, and show similar enhancement factor.

In analogy to the experiments carried out for the photoactive protein, PCP, in the case of the up-converting NCs, we have also performed time-resolved luminescence measurements in order to demonstrate the influence of metallic nanostructures on the radiative processes. For the excitation, an electrically modulated laser operating at 980 nm generating pulses of about 1 µs in width was used. A typical transient collected from NCs deposited onto the silver stripes is presented in Figure 4b. While, similarly to the reference, the decay features a bi-exponential character, the global average decay time was found to be considerably shorter, namely 60 µs. We attribute the reduction of the decay constant to the increase of the emission rate due to the Purcell effect. Two-fold increase of the emission rate is not sufficient to account for the observed enhancement factors; thus, we can conclude that both processes are responsible to a similar degree for the enhancement of the emission intensity of up-converting NCs deposited over the silver stripes.

One of the key advantages of the photochemical approach described in this work concerns the ability of printing an arbitrarily shaped plasmonically active pattern. In Figure 5c, we present a luminescence image measured for up-converting NCs deposited over a 60 µm high symbol of the Dirac constant. The structure was prepared within a couple of minutes with the exposure time of 100 ms for each point. The emission was collected for the 660 nm band of the NCs upon excitation at 980 nm. In accordance with the previous results, the emission of the NCs is also strongly enhanced due to the coupling with metallic nanostructures. Indeed, the enhancement factors amount to approximately 5.

## 3. Materials and Methods

Plasmonically active metallic nanostructures have been prepared on glass substrates using a confocal fluorescence microscope (Eclipse Ti-S, Nikon, Tokyo, Japan), equipped with a continuous-wave laser operating at 405 nm. A high numerical aperture oil immersion objective (Plan Apo 60× NA = 1.4, Nikon, Tokyo, Japan) ensured a small, diffraction-limited diameter of the laser spot (~200 nm) for precise spatial localization of the deposited material. Subsequently, a glass coverslip (#1, Carl Roth, Neunkirchen, Germany), previously cleaned in a Hellmanex II (Hellma, Müllheim, Germany) solution, was mounted on an XY piezoelectric stage of the microscope. Then, 10 µL of a freshly prepared 2 mmol solution of silver nitrate (AgNO_3_, >99%, Sigma-Aldrich, St. Louis, MO, USA) mixed in 1:1 proportion with 2 mmol of trisodium citrate (NaCit, >99%, Acros, Merelbeke, Belgium) was placed on top of the coverslip (Figure 6). The laser, providing an optical power of about 1 mW, which was sufficient to activate the photochemical reaction, was illuminating the sample while moving the piezoelectric stage. The duration of the reaction was controlled by the laser operation time, triggered and synchronized with the movement of the piezoelectric stage by a computer. After the photochemical deposition of silver was completed, the substrate was rinsed in distilled water and dried under ambient condition. In order to avoid silver oxidation, the substrates were used within 24 h from preparation. 

Substrates fabricated using the photochemical approach were characterized with wide-field microscopy and scanning electron microscopy (SEM). In Figure 7, we show a wide-field microscopy transmission image, where silver stripes grouped in four areas are visible. The stripes within each area were obtained with varied exposure time per step during the fabrication. The exposure time for the first stripe was 50 ms per step of the piezoelectric stage (s = 200 nm), increasing by 50 ms for each subsequent stripe. 

On the other hand, the areas were obtained with a different delay time following the movement of the stage. This parameter, defined as delay time, was equal to 10 ms for the first area, 50 ms for the second, and 100 and 150 ms for the remaining two areas. The optical and SEM images (taken using a FEI Nova NanoSEM 450 scanning electron microscope) reveal the influence of both parameters on the morphology of the stripes: for the shortest time and the lowest excitation power of the laser, we found no formation of a distinguished metallic stripe. In contrast, for all other parameter pairings, a clear line emerged, which we associated with photochemical deposition of silver nanostructures. In order to gain insight into the morphology of the stripes, the same sample was analyzed using SEM, as shown in Figure 7b–d. The stripes, regardless of the particular combination of photodeposition parameters, feature an island-like morphology. With the exception of some defects, the islands are rather uniform, with diameters of a few tens of nanometers. We also observe that our approach yields a relatively well controlled distribution of silver within the stripe, as the achieved width of the stripes was of about 1 μm, and the interface between silver islands and the bare glass substrate is also rather sharp. An increase of the exposure time had little influence on the size of silver islands; however, it resulted in an increase of their surface density. Such dimensions, together with the sharp border, make these silver stripes highly promising in the context of applying them for studying plasmonically enhanced photoluminescence. 

## 4. Conclusions

In this work, we demonstrate that silver islands deposited photochemically on glass substrates exhibit a strong plasmonic activity for both Stokes and anti-Stokes emitters. From the fabrication point of view, using this approach, we can precisely define positions of silver stripes, which are characterized with widths of about 1 micrometer. Importantly, in the case of photoactive proteins and rare-earth doped nanocrystals we find a strong increase of the emission intensity. This effect is attributed to enhanced absorption and emission rates (Purcell effect) of these emitters, as a result of the interaction with silver stripes. By correlating the laser illumination with the position of the piezoelectric stage, a pattern of any arbitrary shape can be obtained from the photochemically deposited silver islands. The strong plasmonic activity, together with the ability to control positioning, renders these unique structures potentially applicable for optoelectronic, photovoltaic, and sensory devices.

## Figures and Tables

**Figure 1 ijms-21-02006-f001:**
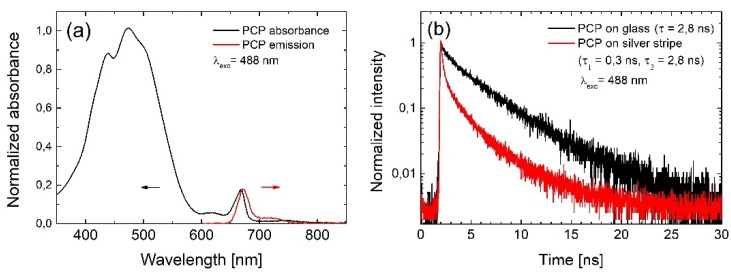
(**a**) Absorption and emission spectra of peridinin-chlorophyll-protein (PCP) complexes. (**b**) Photoluminescence decay transients acquired for PCPs deposited on glass (black line) and silver stripes (red line).

**Figure 2 ijms-21-02006-f002:**
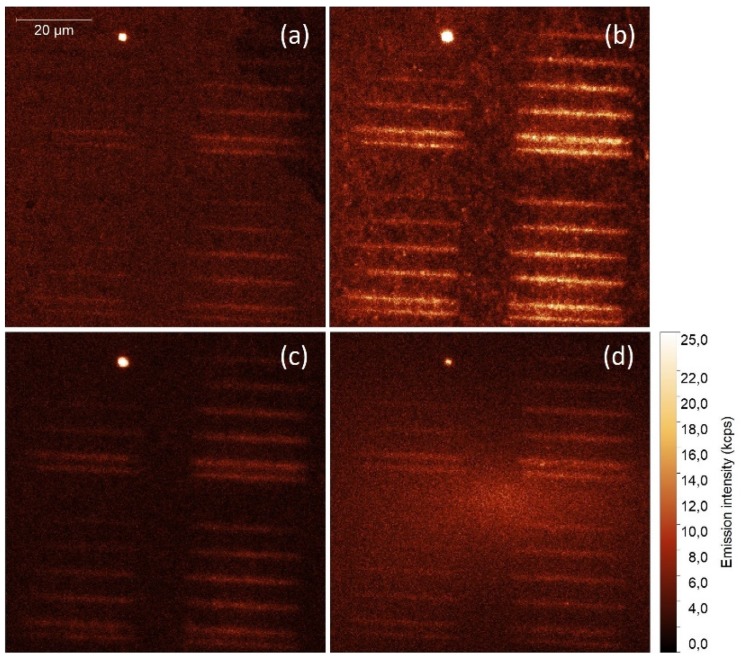
Photoluminescence intensity maps acquired for PCP complexes deposited on a glass substrate with photochemically deposited silver stripes. Excitation wavelengths of: (**a**) 405 nm, (**b**) 480 nm, (**c**) 535 nm, and (**d**) 630 nm were used with the excitation power of 100 μW.

**Figure 3 ijms-21-02006-f003:**
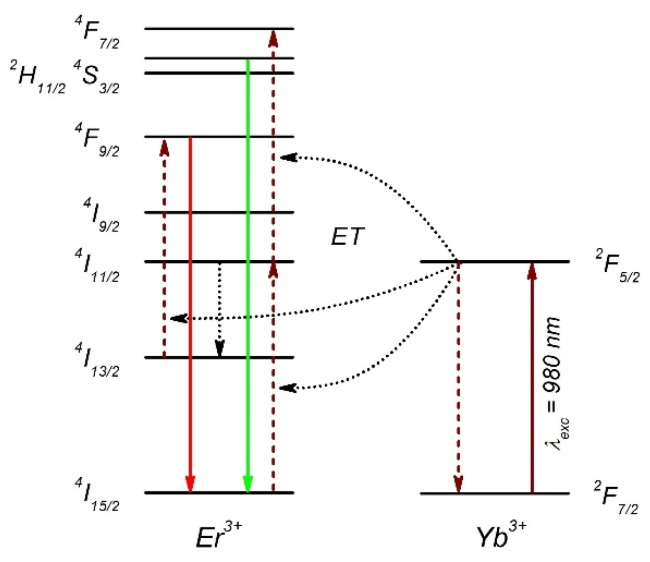
Simplified energy diagram of Er^3+^ and Yb^3+^. Dotted arrows illustrate energy transfer and other non-radiative processes leading to the up-conversion photoluminescence (green and red lines).

**Figure 4 ijms-21-02006-f004:**
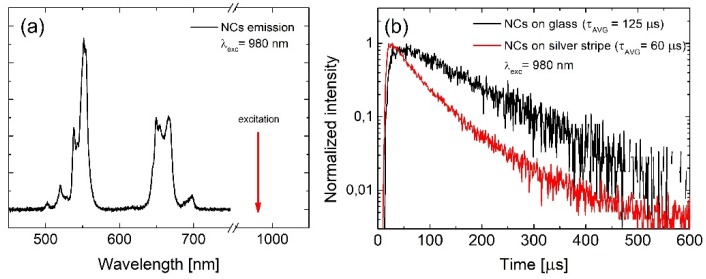
(**a**) Absorption and emission spectra of NaYF_4_:Er^3+^/Yb^3+^ nanocrystals. (**b**) Photoluminescence transients acquired for nanocrystals deposited on glass (black line) and silver stripes (red line).

**Figure 5 ijms-21-02006-f005:**
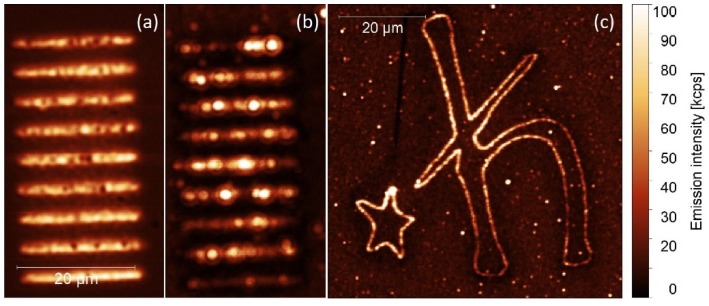
(**a**) Silver paths visualized by a confocal microscope. (**b**) Photoluminescence intensity maps acquired from nanocrystals (NCs) deposited on a silver-decorated glass substrate, and (**c**) from NCs deposited on arbitrarily shaped metallic structures. The excitation wavelength was 980 nm.

**Figure 6 ijms-21-02006-f006:**
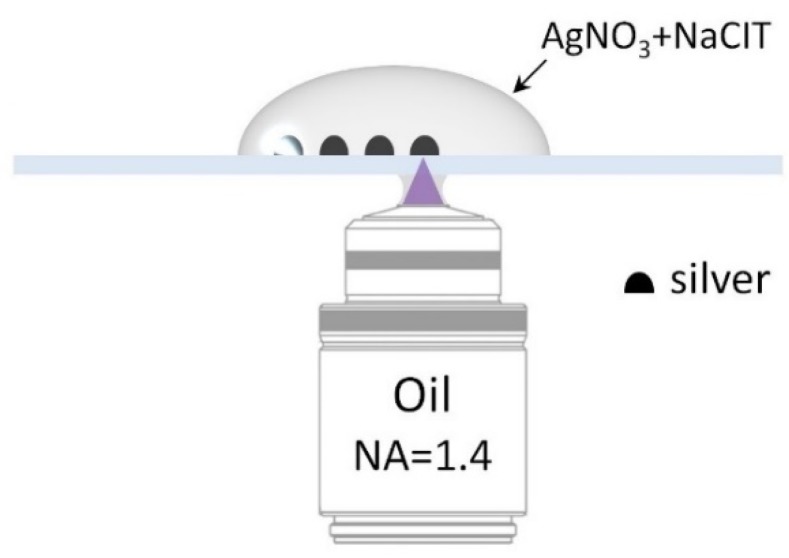
Geometry of the photochemical deposition of silver on a glass substrate. The glass coverslip is mounted above the microscope objective, while a droplet of AgNO_3_ and NaCit is deposited on top. A laser (λ_exc_ = 405 nm) focused on the substrate activates the deposition of silver.

**Figure 7 ijms-21-02006-f007:**
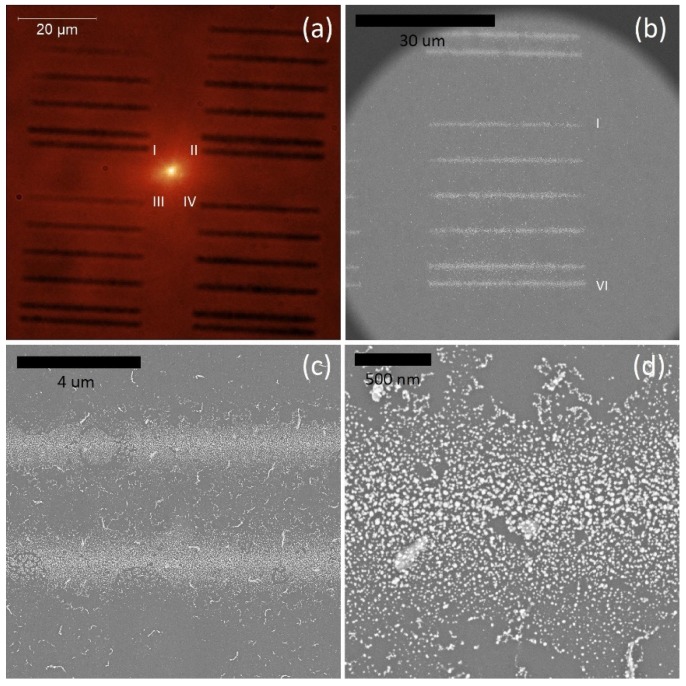
Photochemically synthesized silver islands observed with (**a**) a wide-field optical microscope in transmission mode, and (**b**–**d**) a scanning electron microscope.

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
