# Peer review of "Photochemical Printing of Plasmonically Active Silver Nanostructures"

_ijms, 2020, doi:10.3390/ijms21062006_

Round 1

Reviewer 1 Report

Report on the paper Photochemical Printing of plasmonically active silver nanostructures by M. Czalkowski et al., submitted to Int. J. Molecular Sciences.

In the paper under review, the authors present the use of photochemical reactions, effected using sample illumination with high NA oil - immersed microobjective at 405 nm, to make nanolocal silver metal structures, and the studies of photoluminescence and up-conversion enhanced by these structures. The results are interesting, the fig. 7b is impressive, the paper is well written. I recommend acceptance with a few small comments.

The only one truly important comment is the following. This is known that all silver-made nanostructures in air are subject to rather rapid degradation. What is the experimental situation with the photochemically printed structures discussed in the paper?

Other comments are truly minor.

Line 68. I believe the authors speak abour continuous working laser rather than continuous-wave, as written.

Line 72. Some reference what Hellmanex II solution is should be given.

Line 74. The meaning of the chemical described as “sodium citrate” should be clarified: it is known that three different chemical compositions monosodium citrate, disodium citrate and trisodium citrate  might be named like this.

Lines 127 and 193. When speaking abour spin coating technique, at least the number of rotations per minutes and concentration of active components should be indicated.

Author Response

We thank the Referee for raising a number of important points. We have addressed all of them and marked relevant corrections in the revised version of the manuscript (using a “Track Changes” option). Below is our detailed response to the points raised in the review.

Response to Reviewer 1 comments

Point 1: The only one truly important comment is the following. This is known that all silver-made nanostructures in air are subject to rather rapid degradation. What is the experimental situation with the photochemically printed structures discussed in the paper?

Response 1: Degradation is indeed an important limitation of using silver nanostructures. Therefore, the fabricated substrates were used within 24 hours from preparation. Within this time no measurable differences in observed effects were observed. We added appropriate comments in the manuscript.

Point 2: Line 68. I believe the authors speak about continuous working laser rather than continuous-wave, as written.

Response 2: For silver deposition we used continuous-wave laser, meaning that the beam was turned on all the time, with no impulses.

Point 3: Some reference what Hellmanex II solution is should be given.

Response 3: We clarified this point in the manuscript.

Point 4: The meaning of the chemical described as “sodium citrate” should be clarified: it is known that three different chemical compositions monosodium citrate, disodium citrate and trisodium citrate  might be named like this.

Response 4: We included the information of using specifically trisodium citrate in our experiments.

Point 5: Lines 127 and 193. When speaking about spin coating technique, at least the number of rotations per minutes and concentration of active components should be indicated.

Response 5: We added requested information in the manuscript.

Reviewer 2 Report

In this article, M. Szalkowski et al., explored the strong plasmonic activity for both Stokes- and anti-Stokes emitters based on silver islands photochemically deposited. On one hand, this deposition method is very precise and highly efficient. On the other hand, a strong Purcell effect can be observed from the emissions of the proteins and rare-earth interacting with silver stripes.

The manuscript is well written and clearly to follow, and the results are significant and interesting. Therefore, it deserves to be published in IJMS. Still, please address the following question:

It is better to show the plasmonic resonance of the sliver structures for the demonstration of plasmonic effect for the emission and also Purcell effect.

Author Response

We thank the Referee for raising a number of important points. We have addressed all of them and marked relevant corrections in the revised version of the manuscript (using a “Track Changes” option). Below is our detailed response to the points raised in the review.

Point 1: It is better to show the plasmonic resonance of the sliver structures for the demonstration of plasmonic effect for the emission and also Purcell effect.

Response 1: While we agree with this comment, the stripes fabricated using the photochemical deposition are very thin and cover small fraction of the surface. It is therefore not feasible to detect absorption spectrum with sufficient contrast. Instead, by using several excitation wavelengths in the fluorescence microscopy experiment, we have clearly demonstrated strong wavelength dependence of the emission intensity enhancement. This proves the emergence of the plasmon resonance in the silver stripes at around 480 nm, which is comparable to the values observed for chemically synthesized silver island films (SIFs).